# Validation of the Malay Self-Report Quick Inventory of Depressive Symptomatology in a Malaysian Sample

**DOI:** 10.3390/ijerph19052801

**Published:** 2022-02-28

**Authors:** Lai Fong Chan, Choon Leng Eu, Seng Fah Tong, Song Jie Chin, Shalisah Sharip, Yee Chin Chai, Jiann Lin Loo, Nurul Ain Mohamad Kamal, Jo Aan Goon, Raynuha Mahadevan, Chian Yong Liu, Chih Nie Yeoh, Tuti Iryani Mohd Daud

**Affiliations:** 1Department of Psychiatry, Faculty of Medicine, National University of Malaysia, Jalan Yaacob Latif, Bandar Tun Razak, Cheras, Kuala Lumpur 56000, Malaysia; songjiemailbox@gmail.com (S.J.C.); shalisah@ppukm.ukm.edu.my (S.S.); raynuha@ppukm.ukm.edu.my (R.M.); tutimd@ppukm.ukm.edu.my (T.I.M.D.); 2Department of Psychiatry, Hospital Canselor Tuanku Muhriz, National University of Malaysia, Kuala Lumpur 56000, Malaysia; ecl@ppukm.ukm.edu.my (C.L.E.); nurulain1204@yahoo.com (N.A.M.K.); 3Department of Family Medicine, Faculty of Medicine, National University of Malaysia, Kuala Lumpur 56000, Malaysia; tsf@ppukm.ukm.edu.my; 4Department of Psychiatry, Hospital Permai Johor Bahru, Ministry of Health, Johor Bahru 81200, Malaysia; yeeinchai@yahoo.com; 5Betsi Cadwaladr University Health Board, Wrexham LL13 7TD, UK; jiannlinloo@gmail.com; 6Department of Biochemistry, Faculty of Medicine, National University of Malaysia, Kuala Lumpur 56000, Malaysia; joaan@ukm.edu.my; 7Department of Anaesthesiology and Intensive Care, Faculty of Medicine, National University of Malaysia, Kuala Lumpur 56000, Malaysia; liu.chian.yong@ppukm.ukm.edu.my (C.Y.L.); ychihnie@gmail.com (C.N.Y.)

**Keywords:** validation, reliability, Self-Report Quick Inventory of Depressive Symptomatology (QIDS-SR16), Malaysian, major depressive episode, bipolar depression

## Abstract

Depression is ranked as the second-leading cause for years lived with disability worldwide. Objective monitoring with a standardized scale for depressive symptoms can improve treatment outcomes. This study evaluates the construct and concurrent validity of the Malay Self-Report Quick Inventory of Depressive Symptomatology (QIDS-SR16) among Malaysian clinical and community samples. This cross-sectional study was based on 277 participants, i.e., patients with current major depressive episode (MDE), *n* = 104, and participants without current MDE, *n* = 173. Participants answered the Malay QIDS-SR16 and were administered the validated Malay Mini-International Neuropsychiatric Interview (MINI) for DSM-IV-TR. Factor analysis was used to determine construct validity, alpha statistic for internal consistency, and receiver operating characteristic (ROC) analysis for concurrent validity with MINI to determine the optimal threshold to identify MDE. Data analysis provided evidence for the unidimensionality of the Malay QIDS-SR16 with good internal consistency (Cronbach’s α = 0.88). Based on ROC analysis, the questionnaire demonstrated good validity with a robust area under the curve of 0.916 (*p* < 0.000, 95% CI 0.884–0.948). A cut-off score of nine provided the best balance between sensitivity (88.5%) and specificity (83.2%). The Malay QIDS-SR16 is a reliable and valid instrument for identifying MDE in unipolar or bipolar depression.

## 1. Introduction

Depression is ranked as the second-leading cause of years lived with disability worldwide in 2019 [1]. The COVID-19 pandemic has led to a significant increase of 27.6% in depressive disorders globally [2]. According to the World Health Organization, 48% of the world’s population with depression live in the Southeast Asia and the Western Pacific region [3]. A greater burden of depression in low- and middle-income countries (LMICs) compared to high-income countries (HICs) is suggested by the higher prevalence of major depressive disorder (MDD) in developing regions compared to developed regions [4]. Evidence-based treatment of depression includes measurement-based care such as systematic monitoring of depressive symptoms with a standardized scale, as this has been shown to improve treatment outcomes [5]. Rating scales that have been psychometrically validated in culturally diverse populations are essential in accurately identifying as well as monitoring symptoms and treatment of depression. Characteristics of a valid and practical depression rating scale include brevity, clinician and client user friendliness (self-administered, ease of scoring, etc.), and cost-effectiveness (public-domain accessibility) [5].

The Self-Report Quick Inventory of Depressive Symptomatology (QIDS-SR16) is a 16-item, self-administered questionnaire that assesses the severity of depressive symptoms over the past 7 days [6]. The administration time is approximately 6 min. QIDS-SR16 was shown to have the best discriminatory performance based on diagnostic odds ratio for depression compared to other self-reported depression rating scales such as the Patient Health Questionnaire-9 (PHQ-9), the Hospital Anxiety and Depression Scale (HADS), and Beck Depression Inventory (BDI-II) in a primary care Scottish sample [7]. In addition, QIDS-SR16 is a more detailed and comprehensive depression rating scale that is often preferred over shorter screeners such as PHQ-9, for monitoring treatment response in routine mental health specialty practice or clinical trial settings [8], or HADS which requires license purchasing [9].

A systematic review and meta-analysis by Reilly et al. highlighted the need for more research on the discriminant and convergent validity of QIDS-SR16 beyond the Western world [10]. The QIDS-SR16 has been translated into 38 languages and is accessible from the “Mapi Research Trust” site [11]. The original QIDS-SR16 in English has been validated in a multi-ethnic and multicultural, high-income Asian primary care sample [8]. Nevertheless, there is still a need to validate the translated version of QIDS-SR16 in the local language of LMIC populations.

Malaysia is an upper middle-income country with 32.67 million population with 50.1% of Malaysians being of Malay ethnicity in which their native language is Malay [12]. Malay language is used as the medium for primary and secondary education in Malaysian national schools [13]. The official language is the Malaysian language—Malay. Furthermore, the Malay language is used by 60.5 million people from 13 different countries around the world [13]. The Malay language was adopted as the official language in several countries in Southeast Asia including Indonesia, Malaysia, Singapore, and Brunei. Malaysia and Indonesia both are classified as LMIC [14,15].

A few of the aforementioned questionnaires measuring depression have been translated into Malay. A systematic review by Ali et al. (2016) recommended the use of the Hospital Anxiety and Depression Scale (HADS) and the Patient Health Questionnaire-9 (PHQ-9) as validated depression rating scales in LMIC settings [16]. Nevertheless, these Malay translations of commonly used public-domain depression screening instruments lack a validated cut-off score for a diagnosis of major depressive episode (Depression, Anxiety and Stress Scale; DASS). The incomparability of the depression level scoring between the original 21-item Beck Depression Inventory (BDI) and the construct-validated Malay 20-item BDI is another significant limitation. Given that the Malay PHQ-9 has been validated mainly in Malaysian primary care populations [17], limitations exist in terms of its utility in routine clinical monitoring of depression treatment response in Malaysian mental health specialty settings. Because of the advantages of QIDS-SR16 explained above, translating and validating it in a local setting are necessary.

To the best of our knowledge, there has been no published validation study on the Malay version of the QIDS-SR16. Therefore, the objective of this study was to validate the Malay QIDS-SR16 in identifying the threshold for a current major depressive episode in a culturally diverse Malaysian sample in clinical and community settings. The following psychometric properties of the Malay QIDS-SR16 will be determined: (i) factor analysis to determine the construct validity; (ii) alpha statistic for internal consistency; (iii) receiver operating characteristic (ROC) analysis for concurrent validity with the Malay version of the Mini International Neuropsychiatric Interview (MINI 6.0).

## 2. Materials and Methods

### 2.1. Participants and Procedure

This paper was based on a cross-sectional sub-population analysis of 277 participants where data from diagnostic assessments with the Malay QIDS-SR16 and the Malay MINI 6.0 were available. Participants were selected from 2 larger studies of mood disorder psychiatric patient cohorts (*N* = 418) fulfilling the criteria of healthy volunteers and patients with unipolar or bipolar disorder (lifetime or current major depressive episode). The cohort was consecutively recruited from three urban public hospitals in Malaysia (teaching, general, and psychiatric hospitals). The methodology of the above studies have been described in detail by Chai et al. (2018) and Loo et al. (2021) [18,19]. The study period was from July 2016 to March 2018. The inclusion criteria of the cohort were any Malaysian aged 18 who was able to comprehend the Malay language and gave written informed consent. Participants with major depression would need to fulfil the criteria for a major depressive episode (lifetime or current) according to the Diagnostic and Statistical Manual of Mental Disorders, Fourth Edition, Text Revision (DSM-IV-TR). Current psychosis, mania, hypomania, delirium, or major neurocognitive disorder were excluded. Healthy participants were defined as healthy volunteers with no reported history of psychiatric disorder and did not have major depression (lifetime or current) assessed with the Malay version of MINI 6.0. Prior to screening for study eligibility, study participants received written information regarding the purpose of the study and data anonymization. Study participation was voluntary, and participants were allowed to withdraw from the study at any time without provision of any reason. Study participants with significant depressive symptoms requiring clinical attention were offered resources on help-seeking pathways for mental health services. Ethics approval was granted by the National University of Malaysia Ethics Committee (FF2014-232, GUP-2014-048, and FF-2016-290) and the Medical Research and Ethics Committee, (MREC: NMRR 16-783-29467).

### 2.2. Measures

Study participants completed a demographic questionnaire and the Malay QIDS-SR16. They were later interviewed by a clinician using the Malay MINI 6.0 individually 5–10 min after completing the questionnaire. The clinician was blinded to the result of the Malay QIDS-SR16 prior to MINI 6.0 assessment. Determining if there were missing data on the questionnaire was conducted after MINI 6.0 assessment.

#### Study Instruments

The QIDS-SR16 consists of 16 items each with a Likert scale of 0–3. The QIDS-SR16 score was calculated by summing up the highest response score in each of the three sets from 10 items, measuring (1) sleep disturbances, (2) appetite/weight changes and (3) psychomotor agitation/retardation, and scores in each of the remaining 6 items measuring (1) sad mood, (2) interest, (3) energy/fatigue, (4) self-criticism, (5) concentration, and (6) suicidal ideation.

The total score ranges from 0–27 [6,7]. It assesses the nine DSM-IV-TR criterion symptom domains of major depressive disorder (MDD), which is the same as in later DSM-5 [20,21]. Based on the recommended thresholds to estimate depression severity [6], a score of 0–5 indicates “no depression”; 6–10 “mild depression”; 11–15 “moderately depressed”; 16–20 “severely depressed”; 21–27 “very severely depressed” [6].

Permission to translate and to use the English version of QIDS-SR16 was obtained from the copyright holder, University of Texas Southwestern Medical Center. Translation was performed using the 2 forward and 2 backward-translation techniques by following standard process of translation and adaptation of instruments recommended by World Health Organization (WHO) [22]. All team members involved in this process were bilingual in English and Malay. The forward translation from English to Malay was conducted independently by a master-level clinical psychologist and a biomedical science (MSc) research officer with established research experience in reliability and validity of psychological rating scales. Both forward versions were combined, and the content validity was assessed by a consultant psychiatrist and a doctorate-level clinical psychologist. The combined version was then back-translated into English by 2 medical undergraduates with training in research including translation of rating scales in psychiatry. The 2 English back-translated versions were then combined by another consultant psychiatrist and, subsequently, the content validity was assessed again by the doctorate-level clinical psychologist and the 2 consultant psychiatrists. Face validation of the Malay QIDS-SR16 was performed in 10 patients with MDE who could comprehend in Malay from the tertiary university teaching hospital.

MINI 6.0 is a short-structured diagnostic interview for DSM-IV-TR and ICD-10 psychiatric disorders. Malay MINI 6.0 was validated by Mukhtar et al. (2012) with satisfactory inter-rater reliability (0.67–0.85) and good kappa value (>0.88) [23]. Symptoms of MDE and mania/hypomania were assessed by a clinician using the Malay MINI 6.0. Four clinicians were involved in the assessment. They were all trained in the original MINI 6.0 simultaneously by a single consultant psychiatrist. A categorical diagnosis of a major depressive episode was made based on the interview algorithm of MINI 6.0. Inter-rater reliability was calculated using the Statistical Program for the Social Sciences (SPSS) “KALPHA macro”, according to Krippendorff’s alpha reliability estimate, and produced good agreement across raters (Krippendorff’s α = 0.904, 95% CI 0.811–0.979) [24].

### 2.3. Data Analysis and Estimated Sample Size

Statistical analyses were performed using SPSS version 20 (IBM, Armonk, New York, NY, USA) Data screening was conducted with descriptive statistics. Discriminative validity was calculated using independent *t*-tests comparing patients with current MDE and patients with lifetime MDE in remission with healthy subjects to discriminate between current MDE and without current MDE. Construct validity was examined using the exploratory factor analysis (EFA) to test the underlying dimensionality of the Malay QIDS-SR16. Principal Axis Factoring (PAF) with ProMax rotation was selected. Principle component analysis (PCA) was used to compare both extraction methods [25]. Factorability was inspected in correlation matrix for correlation coefficients over 0.30 [26]. The Keiser–Meyer–Oklin (KMO) test and the Bartlett’s test were examined for feasibility of factor analysis. The KMO test of sampling adequacy is a measure of the shared variance in the items. A KMO value of 0.7–0.79 is interpreted as “middling”, 0.8–0.89 as “meritorious”, and 0.9–1.0 as “marvelous” [27]. A significant Bartlett’s test of sphericity (*p* < 0.05) indicates suitability of the data for factorial analysis [27]. Cronbach’s α was used to evaluate the internal consistency of the questionnaire. The acceptable value of α ranged from 0.70 to 0.95 [28]. For concurrent validity, correlation between QIDS-SR16 and MINI 6.0 was calculated. A receiver operating characteristic (ROC) curve was plotted using the Malaysian version of the MINI 6.0 for presence of current MDE as the “gold standard” to determine optimal cut-off for perceived major depressive episode for the Malay QIDS-SR16. Power calculation for the study was performed with sensitivity (SN) and specificity (SP) set at 90%, with an accuracy (W) of 0.05 and prevalence (P) of 50% [29]. To achieve this power of the study, a total sample size of 277 with 139 being the minimum number of participants without a current major depressive episode was required [30].

## 3. Results

### 3.1. Sample Characteristics

Two hundred and ninety-eight potentially eligible candidates met the study’s inclusion criteria. However, three persons did not give consent to participate in the study, while nine others did not have adequate comprehension of the Malay language. The remaining 286 participants comprised 227 (79.1%) psychiatric patients and 59 (20.9%) healthy participants. One hundred and eighty-one (80.0%) patients were diagnosed with unipolar depressive disorder and 46 (20.0%) with bipolar disorders with current or lifetime major depressive episode. The 59 healthy participants consisted of 21 (35.6%) patient caregivers and 38 (64.4%) volunteers who responded to the study advertisement disseminated to the general public via the study researchers’ networks. Nine (3.1%) participants had missing data in the QIDS-SR16 questionnaire and were excluded from the final validation study analyses. A total of 277 participants were included in the final analyses which consisted of 104 (37.5%) patients with current major depressive episode (MDE) and 173 (62.5%) participants without current MDE (patients with lifetime major depressive disorder in remission and healthy participants). With regards to participants without current MDE, 119 (43.0%) had lifetime MDE in remission and 54 (19.5%) were healthy participants (Table 1). One hundred and seventy-two participants (62.7%) were women. The mean sample age was 43.6 ± 16.3 years. The ethnic composition of the study sample, Malay (54.2%), Chinese (35.3%), Indian (7.2%) and other races (3.3%), closely resembled the Malaysian general distribution [12]. The gender composition among 3 groups of participants closely resembled each other. Patients with current MDE, patients with lifetime MDE in remission and healthy participants consist of 64.1%, 62.4%, and 61.1% of female respectively with mean age of 44.9, 46.2, and 35.4 respectively. The majority of patients with current MDE (53.8%) and patients with lifetime MDE in remission (61.5%) were of Malay ethnicity, while 42.6% of the healthy participants were of Chinese ethnicity. Fifty-point three percent of the participants were married. About four-fifths (79.6%) of the healthy participants had higher education qualification, while patients with current MDE and patients with lifetime MDE in remission who had higher education qualification were 43.8% and 55.2%, respectively. The percentage for employment status for patients with current MDE and patients with lifetime MDE in remission closely resembled each other. The majority of healthy participants (75.9%) were on full-time employment.

### 3.2. Discriminative Validity

The mean score of QIDS-SR16 for patients with current MDE was significantly higher than the healthy group (Table 2), whereas the mean score for patients with lifetime MDE, currently in remission, was comparable with the mean score of the healthy group (Table 2). This indicates that when patients are in remission, the severity of their depressive symptoms could return to the general healthy baseline. The Malay QIDS-SR16 demonstrated the ability to discriminate between current MDE and without current MDE.

### 3.3. Construct Validity

Two hundred and seventy-seven subjects were included in the EFA. All correlation coefficients among the items were over 0.30 indicating that individual items accounted for at least 30% of the relationship within the data, which allowed for meaningful factor analysis (Table 3). The KMO Test was excellent (0.921) with significant Bartlett’s Test (*p* < 0.001) indicating adequacy and suitability of the data for factorial analyses. Sample size sufficiency was evidenced from the stability of the solution pointed out above [29]. The sample to variance (STV) ratio was 30:1. A single factor was extracted by using both PAF and PCA. Forty-five-point seven percent of the total variance was explained by using PAF and 51.5% by PCA. The explained variance is commonly low (50–60%) in humanities studies [25].

### 3.4. Reliability Analysis

The Cronbach’s α of the QIDS-SR16 in the total sample (*n* = 277) was 0.88. This indicated adequate internal consistency based on the acceptable range for Cronbach’s α from 0.70 to 0.95 [28].

### 3.5. Concurrent Validity

MINI 6.0 diagnosis for current MDE was used as the criterion standard. Based on the ROC analysis with the reference criterion standard, the QIDS-SR16 (Malay version) provided evidence for good validity with robust area under the curve of 0.916 (*p* < 0.000, 95% CI 0.884–0.948). The optimal cut-off for Malay QIDS-SR16 was 9 points with a sensitivity of 88.5% and specificity of 83.2%. With a lower cut-off at 7 points, the sensitivity was 96.2% and specificity was of 70.5%, providing a higher yield in detecting MDE (Figure 1, Table 4) with an expense of higher false positive rate.

Using a cut-off score of 9 points, the results demonstrated a positive likelihood ratio (LR+) of 5.3 and strong negative likelihood ratio (LR−) of 0.14. This indicates that a positive test increases the likelihood of disease by five times in comparison to their pre-test odds. Similarly, a negative test decreases their likelihood for disease to 0.14 of their pre-test odds. Therefore, based on a 12% prevalence of MDD in Malaysia, a positive test increases the odds for MDE to 60%, and a negative test reduces the odds to only 1.68% [31].

## 4. Discussion

This was a cross-sectional study with the goal of evaluating the psychometric properties of the Malay QIDS-SR16 among a Malaysian sample. Healthy participants and patients with lifetime or current unipolar or bipolar MDE were included. The targeted sample size of 277 was achieved to ensure adequate power for construct and concurrent validity. The used of validated Malay version of MINI 6.0 increased the robustness of the study.

Unidimensionality was established evidenced by extraction of a single factor with good factor loadings. Three domains (i.e., sad mood, reduced concentration and reduced interest) had the highest factor loading. This finding was consistent with clinical understanding of MDE in which low mood and anhedonia were well recognized as core symptoms of depression. Reduced concentration explained well the significance of cognitive impairment among depressed patients. McIntyre et al. (2013) concluded that cognitive deficits in depression were significant and a principal mediator of psychosocial impairment, notably workforce performance [32]. The factor loading for sleep disturbances was still within the acceptable range of >0.4 [33]. However, sleep disturbances contributed the least to depression in the analyses. This could be explained by the high prevalence of insomnia among the Malaysian population. Zailinawati et al. in 2008 and 2012 reported a high prevalence of insomnia among the Malaysian urban community (33.8%) and primary care patients (60%) [34,35]. These authors reported that insomnia was significantly associated daytime dysfunction and depression. Limitations of our cross-sectional study design could be addressed in future locally contextualized prospective studies in order to elucidate the relationship (cause and effect) between primary insomnia disorders, major depression, and the impact on functionality as potential targets for early intervention to improve clinical prognosis.

The Malay QIDS-SR16 showed good internal consistency (Cronbach’s α of 0.88) in line with studies by Trivedi et al. (2004), Doraiswamy et al. (2010), Bernstein et al. (2010), Cameron et al. (2013), and Sung et al. (2013) which yielded similar Cronbach’s α values of 0.86, 0.85, 0.80, 0.85–0.89, 0.88, and 0.87, respectively [6,8,36,37,38,39]. It should be noted that the reliability of the QIDS-SR16 were not affected despite its use across wide spectrum of cultural and language differences.

Receiver operating characteristic analysis suggested a cut-off at 9 points to achieve an optimal balance between SN (88.5%) and SP (83.2%), which served as a reference in screening for patients meeting diagnostic criteria for MDE in unipolar or bipolar depression. This suggested cut-off is higher than the original study by Rush et al. (2003) in which 6 points was considered as mild clinical depression (SN = 79%, SP = 81%) [6]. The study was conducted among adult outpatients (*N* = 596) in 12 US academic centers, who were treated for chronic nonpsychotic MDD. Hamilton Rating Scale for Depression (HAM-D) was used as criterion standard [6]. Findings from study by Liu et al. (2014) who conducted their study among China patients with MDD (*N* = 1164) was in agreement with Rush et al. in setting 6 points as depression severity thresholds [40]. Montgomery-Asberg Depression Rating Scale (MADRS) was used as the criterion standard [32]. On the other hand, Lamoureux et al. (2010), who conducted their study among primary care medical patients (*N* = 155) in an urban Midwestern US city, yielded higher cut-off scores of 13 and 14 points (SN = 76.5%, SP = 81.8%,) [41]. Structured Clinical Interview for DSM-IV-TR Axis I Disorders (SCID) was selected as criterion standard [41]. Study conducted by Sung et al. (2013) among Singapore primary care patients (*N* = 400) had identified 9 points as the optimal cut-off for MDD (SN = 83.3%, SP = 84.7%, LR+ = 5.4, LR− = 0.20), which was identical to the Malaysian study [8]. Both Singapore and Malaysian studies used MINI as a criterion standard. Various cut-off scores in these studies reflected true differences between the sample populations and, thus, highlighted the importance of local validation study for measuring instruments. These differences were subjected to the selection of sample population (primary care patients or patients with MDD) and criterion standard for concurrent validity. The use of clinician-rated screening tool for depression such as HAM-D and MADRS provided score in the form of continuous data within a set interval [42]. Semi-structured clinician-rated diagnostic tools such as SCID and MINI provided dichotomy nominal data of either “yes” and “no” for a current MDE [43,44]. Based on the above observation, studies among a more heterogeneous sample group using dichotomy diagnostic tools as a criterion standard seemed to produce a higher cut-off score. This could lead to the inference that higher QIDS-SR16 cut-offs in these studies were probably a reference in screening for patients meeting diagnostic criteria for a current major depressive episode (MDE).

Sung et al. had identified another cut-off at 7 point (SN = 94.4, SP = 77.9, LR+ = 4.4, LR− = 0.07) for minor depressive disorder, which was defined as depression fulfilling 2 to 4 out of 9 symptoms with at least a symptom being depressed mood or loss of interest [8]. Minor depressive disorder was listed in Appendix B of the DSM-IV-TR (criteria sets and axes provided for further study) but was later removed in DSM-5 [20,21]. In the Malaysian study, a cut-off at 7 points with good SN (96.2%) and acceptable SP (70.5%), with moderate LR+ (3.3) and good LR− (0.05) were considered to be used as a screening reference for sub-syndromal depression. Integrating the result of both studies conducted in the same South East Asia region, this value could serve as a reference for screening for sub-syndromal depression.

### 4.1. Strengths and Limitations

To the best of our knowledge, this is the first published validation study for the Malay QIDS-SR16 in Malaysia, a culturally diverse LMIC. The external validity of the study was potentially compromised by the limitation of proper random sampling, predisposing the study to recruitment biases.

### 4.2. Implications

The implications for this study include enabling mental health clinicians and researchers to utilize the validated Malay QIDS-SR16 in identifying clinically significant major depressive episodes and sub-syndromal depression. In addition, the Malay QIDS-SR16 allows clinicians to cost-effectively implement measurement-based care, as this validated instrument can objectively monitor patient-reported outcomes in treatment of depression [45].

## 5. Conclusions

The Malay QIDS-SR16 is a valid instrument with a unidimensional construct and is a reliable tool with good internal consistency. Future studies are warranted to establish the clinical utility and feasibility of the Malay QIDS-SR16 in wider LMIC settings.

## Figures and Tables

**Figure 1 ijerph-19-02801-f001:**
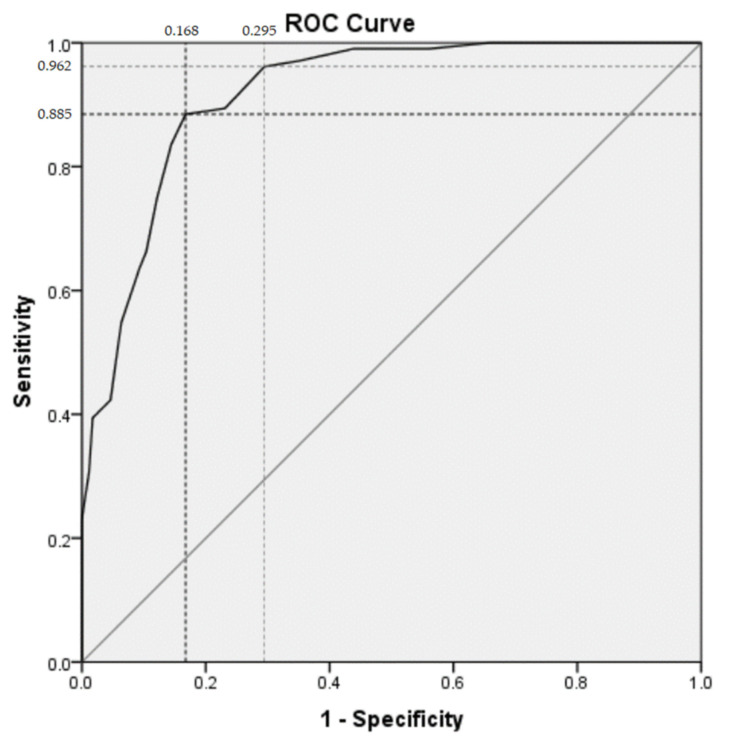
Receiver operator characteristics (ROC) curve showing the probability of predicting a MINI diagnosis of a major depressive episode using the QIDS-SR16 score.

**Table 1 ijerph-19-02801-t001:** Clinical and demographic characteristics of the sample (*N* = 277).

Characteristic	Patients with a Current MDE ^#^	Patients with Lifetime MDE ^#^ in Remission	Healthy Participants	Total
Female (%)	64.1	62.4	61.1	62.7
Age (mean ± SD)	44.9 ± 16.3	46.2 ± 17.4	35.4 ± 10.6	43.6 ± 16.3
Ethnicity (%)				
Malay	53.8	61.5	38.9	54.2
Chinese	34.6	32.5	42.6	35.3
Indian	11.5	5.1	3.7	7.2
Others	0	0.9	14.8	3.3
Marital Status (%)				
Married	48.0	52.9	49.1	50.3
Educational level (%)				
<10 years	6.3	8.6	3.7	6.3
>10 years	50	36.2	16.7	33.8
Higher education	43.8	55.2	79.6	60.0
Employment (%)				
Unemployed or part time	18.0	14.7	7.4	13.4
Fulltime employment	44.0	47.1	75.9	55.2
Retired/pensioner	10.0	16.2	1.9	9.9
Housewife	14.0	7.4	7.4	9.3
Students	14.0	14.7	7.4	12.2
QIDS-SR16 * (Malay version) score (%)				
0–5	1.0	58.0	51.9	35.4
6–10	15.4	23.5	42.6	24.2
11–15	41.3	12.6	3.7	21.7
16–20	23.1	5.9	1.9	11.6
>20	19.2	0	0	7.2

* Self-Report Quick Inventory of Depressive Symptomatology. ^#^ Major depressive episode.

**Table 2 ijerph-19-02801-t002:** QIDS-SR16 mean ± SD for the healthy and patient groups based on MINI 6.0 assessment.

MINI 6.0 Classification	* n * (%)	QIDS-SR16 Score Mean (SD)	*t-*Value	*p*-Value *
Healthy	54 (20.9)	5.80 (2.90)		
Patients with current MDE	104 (36.1)	15.29 (5.01)	15.053	<0.001
Patients with lifetime MDE	119 (43.0)	5.84 (4.94)	0.073	0.942

QIDS-SR16, Self-Report Quick Inventory of Depressive Symptomatology; MINI, Mini-International Neuropsychiatric Interview. * Independent *t*-test comparing with the healthy group.

**Table 3 ijerph-19-02801-t003:** Factor loading for each item/domain in QIDS-SR16.

QIDS-SR16 Items/Domains	Factor Loading
Sad mood (item no. 5)	0.775
Reduced concentration (item no. 10)	0.758
Reduced interest (item no. 13)	0.736
Self-criticism (item no. 11)	0.676
Suicide ideation (item no. 12)	0.664
Psychomotor agitation/retardation (items no. 15–16)	0.663
Reduced energy/fatigue (item no. 14)	0.625
Appetite/weight changes (items no. 6–9)	0.613
Sleep disturbances (items no. 1–4)	0.538

QIDS-SR16, Self-Report Quick Inventory of Depressive Symptomatology.

**Table 4 ijerph-19-02801-t004:** ROC analysis of QIDS-SR16 (Malay version) of different cut-offs for MDE.

Cutoff Score (Points)	Number of Subjects	Sensitivity (%)	Specificity (%)	Positive Likelihood Ratio	Negative Likelihood Ratio	Positive Predictive Value (%)	Negative Predictive Value (%)
≥9	121	88.5	83.2	5.3	0.14	76.0	92.3
≥7	151	96.2	70.5	3.3	0.05	66.2	96.8

ROC, Receiver operator characteristics; QIDS-SR16, Self-Report Quick Inventory of Depressive Symptomatology; MDE, major depressive episode.

## Data Availability

The data that support the findings of this study are available upon request from the corresponding authors. The data are not publicly available due the fact of privacy and/or ethical restrictions.

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
