# Peer review of "Validation of the Malay Self-Report Quick Inventory of Depressive Symptomatology in a Malaysian Sample"

_ijerph, 2022, doi:10.3390/ijerph19052801_

Round 1
Reviewer 1 Report
Comments to the Authors,
The Paper Validation of the Malay Quick Inventory of Depressive Symptomatology-Self Report 16 Items in a Malaysian Sample is an interesting and easy-to-read paper. It is well described and organized. The authors detect a clear need to translate and validate easy-to-use, available questionnaires to assess depression and evaluate interventions in their own language. The introduction is solid and well explained, and it is useful and applicable research.
In terms of contributing some ideas, the procedures are clear but perhaps the authors could explain what the minimum number of controls needed is in the statistics description. It would be also interesting to have some sociodemographic data on the control group in a similar table or a description similar to that of the psychiatric group.
Continuing with the control group, could the authors explain a bit more about the caregivers in the control group and the possible implications for results? Could it have some implications in depression symptomatology or insomnia results?
In the discussion, could the authors expand a little more on the insomnia results and give an explanation of them? Reference 33 mentions the “Prevalence of insomnia…” (Zailinawati et al 2012); it is linked to depression symptomatology and perhaps it would be interesting to discuss in more depth the possible implications, limitations or future research in this aspect.
Reviewer 2 Report
-Paragraph Participants and procedure should be updated with demographic and disease related factors (% of male/female, mean (sd) age, education level etc.) of the sample: healthy participants, patients with current MDE and patients with lifetime MDE.
-Paragraph Data analysis should have explained which test was used to determine differences between groups
-Table 1 should be also divided into three columns: helathy participants (N), patients with current MDE (N) and patients with lifetime MDE (N).
-Figure 1 is not necessary
-Differences in age and sex between groups (ie.healthy and patients with depression) should be reported in section 3.1.
-In Table 2 please report t-value and the level of statistical significance
-The Discussion should be updated to be more clearer. Please delete Sensitivity, Specificity and Lihood ratio values from Discussion.
